# Impact of a *Withania somnifera* and *Bacopa monnieri* Formulation on SH-SY5Y Human Neuroblastoma Cells Metabolism Through NMR Metabolomic

**DOI:** 10.3390/nu16234096

**Published:** 2024-11-28

**Authors:** Maria D’Elia, Carmen Marino, Rita Celano, Enza Napolitano, Anna Maria D’Ursi, Mariateresa Russo, Luca Rastrelli

**Affiliations:** 1Department of Pharmacy, University of Salerno, Via Giovanni Paolo II, 132, 84084 Fisciano, Italy; mdelia@unisa.it (M.D.); cmarino@unisa.it (C.M.); rcelano@unisa.it (R.C.); enapolitano@unisa.it (E.N.); dursi@unisa.it (A.M.D.); 2National Biodiversity Future Center—NBFC, 90133 Palermo, Italy; 3Dipartimento di Scienze della Terra e del Mare, University of Palermo, 90123 Palermo, Italy; 4Department of Agriculture Science, Food Chemistry, Safety and Sensoromic Laboratory (FoCuSS Lab), University of Reggio Calabria, Via dell’Università, 25, 89124 Reggio Calabria, Italy; mariateresa.russo@unirc.it

**Keywords:** *Withania somnifera*, *Bacopa monnieri*, stress, SH-SY5Y cells, UHPLC-HRMS/MS, ^1^H NMR metabolomic

## Abstract

**Objectives**: This study investigates the effectiveness of an herbal formulation, STRESSLESS (ST-65), which combines ashwagandha (*Withania somnifera*) and bacopa (*Bacopa monnieri*), on SH-SY5Y human neuroblastoma cells. Given the rising interest in natural compounds for neuroprotection and stress alleviation, we aimed to explore the cellular and molecular effects of this formulation. **Methods**: Utilizing a nuclear magnetic resonance (NMR) metabolomic approach and ultra-high-performance liquid chromatography-high-resolution mass spectrometry (UHPLC-HRMS), we identified key bioactive compounds in ST-65, including withanolides from ashwagandha and bacosides from bacopa. **Results**: Our findings indicate that ST-65 treatment significantly alters the metabolic profile of SH-SY5Y cells. Key changes included increased levels of metabolites linked to neuroprotection, energy metabolism, and antioxidant defense. Notable enhancements were observed in specific amino acids and neuroprotective compounds, suggesting activation of neuroprotective mechanisms and mitigation of stress-induced damage. **Conclusions**: The study reveals a complex phyto-chemical profile of ST-65 and underscores its potential as a natural active agent for addressing stress-related neurodegenerative conditions. These insights into neuronal mechanisms provide a foundation for further exploration of herbal formulations in neuroprotection.

## 1. Introduction

Stress is a common experience that can have significant effects on both mental and physical health. Stress triggers the release of cortisol, a hormone produced by the adrenal glands. While cortisol helps regulate metabolism, immune response, and blood pressure, chronic stress can lead to prolonged high levels of cortisol, which may negatively affect physical and mental health [1]. Some effects of elevated cortisol include: increased anxiety and depression [2], weight gain [3], sleep issues [4], impaired immune function: chronic stress can weaken the immune response, making you more susceptible to illness [5]. To manage stress and cortisol levels, practices such as regular exercise, mindfulness meditation, adequate sleep and a balanced diet are increasingly considered [6]. Some natural compounds have been studied for their potential to help manage stress and promote relaxation, among them ashwagandha, an adaptogenic herb traditionally used in Ayurvedic medicine, has been reported in many studies for its ability to help the body manage stress, suggesting that it may help lower cortisol levels, reduce anxiety, and improve overall stress resilience [7]. Several clinical studies have investigated the effects of ashwagandha on stress, anxiety, and overall well-being. Ashwagandha may enhance sleep quality [8], which is often disrupted by stress and improve cognitive function and memory, which can be negatively impacted by stress [9]. A systematic review analysed human randomized controlled trials with a treatment arm that included *Withania somnifera* as a remedy for anxiety or stress and concluded that it may improve overall health and well-being, particularly in reducing stress and anxiety [10]. Guo & Rezaei [11] conducted a comprehensive review of the scientific literature focusing on the effects of ashwagandha supplementation, particularly its impact on antioxidant response and athletic performance. Their findings likely highlight how ashwagandha may enhance physical performance by reducing exercise-induced oxidative stress and improving recovery times. A meta-analysis investigating the effects of ashwagandha on physical performance, including fatigue, found that ashwagandha may help alleviate symptoms of fatigue [12]. Additionally, stress, fatigue, and aging are linked to decline in sex hormone levels in both men and women [13]. In a randomized, double-blind, placebo-controlled trial, authors reported that supplementation with 200 mg of ashwagandha extract (twice daily for 12 weeks) in overweight or mildly obese men and women, experiencing high stress and fatigue symptoms, was linked to a significant increase in free testosterone (FT) and luteinizing hormone (LH) levels in men [14]. Sex hormone metabolism, on the other hand, supports neuroprotection by influencing cellular mechanisms that protect against neurodegeneration, inflammation, and oxidative stress [15]. Withanolides are the biologically active compounds of *Withania somnifera*, a group of naturally occurring C28 steroids built on an ergostane skeleton functionalized at carbons 1, 22 and 26 [16].

Multiple studies have shown that withanolides reduces oxidative stress and modulates glutamatergic neurotransmission [17], and possesses neuroprotective properties against neurodegeneration caused by hypobaric hypoxia as well as cerebral ischemia-reperfusion injury [18]. These compounds are also recognized for their potential in treating various neurodegenerative diseases, including Alzheimer’s [19] and Parkinson’s disease [20]. Furthermore, because they are derived from natural plant sources, they have gained widespread popularity as oral supplements worldwide, owing to their powerful therapeutic properties [7].

Many studies have considered *Bacopa monnieri* L. to prevent or treat neurodegenerative conditions [21,22] and for its potential benefits in managing stress [23]. This plant is known as brahmi or “medhya rasayanas” in Ayurveda (meaning brain tonic). Bacopa has been shown to exert antioxidant effects, which may help protect neuronal cells from oxidative damage, a significant factor in neurodegenerative diseases like Alzheimer’s [24] and Parkinson’s [25]. A recent systematic review indicated that Bacopa can improve cognitive function, including memory, attention, and learning, which may be particularly beneficial for individuals at risk of neurodegenerative disorders [26]. Authors included clinical trials and preclinical studies that evaluated the efficacy of Bacopa in improving cognitive symptoms and overall brain health in populations at risk for or diagnosed with neurodegenerative conditions. The herb contains active, dammarane-type triterpenoidal saponins named bacosides, which are believed to promote synaptic communication, reduce inflammation, and enhance neurogenesis [27]. In our clinical experience, Ashwagandha-Bacopa supplementation through the administration of a formulation obtained from the combination of 600 mg dry extract of ashawagandha and 50 mg dry extract of bacopa has been able to alleviate the stress and anxiety experienced by patients during restrictive dietary regimens [28]. In this study, to confirm our clinical observations, we reported the impact on SH-SY5Y human neuroblastoma cells of a stress-reducing formulation consisting in 600 mg of ashwagandha and 50 mg of Bacopa in standardized extracts (named ST-65) using NMR metabolomic. We assessed the efficacy and safety of ST-65 providing valuable insights into metabolic alterations and potential therapeutic effects. We also developed a credible analytical method to identify the chemical components of ST-65. Eighteen chemical constituents, such as derivatives of flavonoids, withanolides bacosides and their derivatives, were identified or tentatively characterized.

## 2. Materials and Methods

### 2.1. Chemical and Standards

Analytical grade ethanol (EtOH), methanol (MeOH), chloroform, MS grade formic acid (HCOOH), and the reference standards of rutin, hesperidin, quercetin, were provided by Merck Chemicals (Milan, Italy). MS grade acetonitrile and water were purchased from Romil (Cambridge, UK). Ultrapure water (18 MΩ) was prepared using a Milli-Q purification system (Millipore, Bedford, TX, USA).

### 2.2. ST-65 Formulation

STRESSLESS (ST-65) formulation consisted of the following extracts: KSM-66 ashwagandha (5% withanolides), a water-based extract of *Withania somnifera* (L.) Dunal roots made by Ixoreal Biomed of Los Angeles, California and a water-based extract of *Bacopa monnieri* (L.) Pennell aerial part (20% bacosides) kindly supplied by Laboratoarele Medica srl (Bucharest, Romania). The extracts were blended at Department of Pharmacy of University of Salerno in a 12 to 1 ratio accurately weigh 1200 mg of Ashwagandha E.S. and 100 mg of Bacopa using an analytical balance. The ingredients were combined and mixed in a mixing bowl thoroughly to ensure even distribution of the extracts.

### 2.3. UHPLC-HRMS/MS Analysis

The samples were dissolved in a hydroalcoholic solution (70% *v*/*v*) with a matrix/solvent ratio of 1:20. The analysis were performed using a Vanquish Flex UFPLC system interfaced to an Orbitrap Exploris 120 mass spectrometer, equipped with a heated electrospray ionization source, HESI-II (Thermo Fisher Scientific, Milano, Italy). A Kinetex C18 column (100 × 2.1 mm I.D., 2.6 µm; Phenomex, Bologna, Italy) was selected to obtain the best efficiency and resolution, at flow rate of 500 µL min^−1^ at 30 °C. The mobile phase was a binary gradient of water (A) an MeCN (B), both containing 0.1%, *v*/*v*, formic acid. The gradient elution program is as follows: 0–3 min, 2% B; 3–5 min, 2–13% B; 5–9 min, 13% B; 9–12 min, 13–18% B; 12–13 min, 18% B; 13–17 min, 18–30% B; 17–20 min, 30% B; 20–30 min, 30–40% B; 30–38 min, 40–60%B; 38–39 min, 60–98% B; after each injection (5 µL), washing (98% B, 4 min) and re-equilibration of the column (2% B, 5 min) were performed. The mass spectrometer was operated in negative and positive mode. A Full MS data dependent MS/MS (Full MS/dd-MS2) acquisition mode was used, withFull MS scan resolution at 60,000 FWHM (scan range *m*/*z* 100–1000), and dd-MS2 scan set at 30,000 FWHM. A stepped collision energy HCD (20, 40 and 60) was applied for fragmentation. Detected compounds were identified to the corresponding spectral characteristics: mass spectra, accurate mass; fragmentation pathways and retention time. Xalibur software (version 4.6) was used for instrument control, data acquisition and data analysis.

### 2.4. Cell Culture

The human neuroblastoma cell line SH-SY5Y was obtained from American Type Culture Collection (ATCC, Rockville, MD, USA). Cells were cultured in Dulbecco’s Modified Eagle Medium (DMEM, 4500 mg/mL glucose) supplemented with 10% (*v*/*v*) fetal bovine serum, 2 mM L-glutamine, 100 U/mL penicillin, and 0.1 mg/mL streptomycin. Cells were grown in 100 mm culture dishes (Corning, Corning, NY, USA) in a humidified environment containing 5% CO_2_ at 37 °C and split every 2 days.

### 2.5. Cell Viability Assay

Cell viability was established by measuring mitochondrial metabolic activity to reduce the tetrazolium salt WST-8 (2-(2-methoxy-4-nitrophenyl)-3-(4-nitrophenyl)-5-(2,4-disulfophenyl)-2H-tetrazolium, monosodic salt) through Cell Counting Kit-8 (CCK-8 Cat. CK04, Dojindo Laboratories, Rockville, MD, USA). Briefly, SH-SY5Y (8 × 10^3^ cells/well) was plated into 96-well plates for 24 h, and then ST-65 in the range of 0.80–100 µg/mL was added for an additional 24 h. At the end of the treatment, CCK-8 solution (10 μL) was added to the cell media and incubated for 1 h in a humidified incubator. Absorbance was detected at a wavelength of 450 nm. Results are expressed as means ±SD of 3 independent experiments performed in triplicate and reported as the percentage of viable cells vs. the untreated control.

### 2.6. Exposure of SH-SY5Y to STRESSLESS (ST-65) Formulation

Cells were pleated in 100 mm culture dishes and allowed to adhere overnight. Afterward, ST-65 (25 μg/mL) was added and incubated for 24 h. For the control group, cells were treated only with a vehicle. At the end of treatments, cell medium was collected, and the dishes were washed with cold PBS (pH 7.4) to remove media residues just before the metabolite extraction procedure.

### 2.7. Sample Extraction

The culture medium was collected from each plate in microcentrifuge tubes and centrifuged at 1000× *g* for 5 min. For extracting cellular metabolites, a dual-phase extraction (methanol:chloroform:water; 1:1:1) was used after cell collection by scraping and homogenization [29]). The two phases were separated after centrifugation at 6000 rpm for 10 min at 4 °C. The resulting polar extracts were dried under vacuum in an SP-Genevac EZ-2 4.0 concentrator, and the lipophilic extracts were dried under a nitrogen flow for future analysis. Both growth media and cell extracts were stored at −80 °C before NMR analysis.

### 2.8. NMR Spectra Acquisition

Bruker Ascend™ 600 MHz spectrometer was used to acquire the spectra. The spectrometer was equipped with a 5 mm triple resonance Z gradient TXI probe (Bruker Co., Rheinstetten, Germany) at 298 K. TopSpin, version 3.2 was used for the spectrometer control and data processing (Bruker Biospin). All the experiments performed as Nuclear Overhauser Enhancement Spectroscopy (NOESY) 1D were acquired in triplicate. Spectra acquisition was made using 12 ppm spectral width, 20 k data points, presaturation during relaxation delay and mixing time for water suppression [30] and spoil gradient, 5 s relaxation delay, and mixing time of 10 ms. A weighted Fourier transform was applied to the time domain data with a line widening of 0.5 Hz followed by a manual step and baseline correction in preparation for targeted profiling analysis.

### 2.9. NMR Spectra Analysis

NMR spectra of SH-SY-5Ycells cultures eso and endometabolome were analysed using an untargeted metabolomic approach. All the spectra were assigned using Chenomx NMR-Suite v8.0 (Chenomx Inc., Edmonton, AB, Canada) and quantified by NMRProcFlow as previously reported [30]. The quantification matrices reported the metabolites identified and quantified in the eso and endometabolome of SH-SY-5Y treated and untreated with ST-65 were analyzed using the open-source tool Metaboanalyst 6.0 [31]. The Volcano plot combined T-test and Fold Change performed the univariate approach [31,32]. After normalization by sum log and Pareto, we applied a supervised multivariate approach partial least-squares discriminant analysis (PLS-DA) method. The reliability of the supervised model was analyzed using a cross-validation approach, considering the accuracy and parameters Q2 and R2. The metabolites responsible for clusters’ separation in the PLS-DA score plot were classified according to VIP, considering only the metabolites with VIP > 1 [33]. Enrichment Pathways tools were applied to identify the dysregulated biochemical pathways by KEGG database. Only the KEGG paths that reported a rate of false discoveries (FDR) lower than 1, the *p*-value lower than 0.05 and the hits value related to the number of metabolites belonging to the pathway > 1, were chosen. [31].

## 3. Results

### 3.1. UHPLC-HRMS/MS Analysis

In the present study, an UHPLC-HRMS/MS method for profiling and characterization of constituents present in the ST-65 formulation was developed. The UHPLC conditions were optimized to obtain maximal chromatographic resolution and MS signal. Figure 1 shows representative chromatogram of ST-65 under optimal conditions. HRMS/MS analysis were performed in positive and negative ionization mode to obtain complementary information needed to characterize ST-65 formulation. To avoid repetitions, data have been reported only in positive ionization mode (Table 1) as they show higher response in the detection. Compound assignments were made by HRMS spectra, tandem mass spectrometry experiments and when possible, by comparing MS spectra and retention time with authentic standards, or with chemo-taxonomic data reported in literature. Eighteen (**1**–**18**) major peaks were detected in ST-65 formulation, and their retention times and MS data are listed in Table 1. UHPLC-HRMS/MS analysis allowed to identify four flavonoids (**3**–**6**), ten withanolides (**7**–**16**) and one bacopa saponin (**17**). Compounds **3**, **5** and **6** were ascertained to be rutin, hesperidin and quercetin, respectively, by reference standards. Compound **4** in MS^2^ spectra showed a base peak at *m*/*z* 287.0548 corresponding to the aglycon kaempferol and the characteristic losses of hexose and deoxyhexose residues. Thus, compound **4** could be tentatively identified as kaempferol-hexose-deoxyhexose. Compounds **7**–**16** were identified as withanolides, 28-carbon steroidal lactones, built up on an intact or rearranged ergostane framework [34]. Their structural chemistry produce limited fragmentation in ESI mass spectra. Therefore, in MS^2^ spectra, only losses of the sugar units linked to the aglycone were detected. Thus, according MS analysis and literature data [35] the compounds **7**–**15** could be identified as withanolides reported in Table 1. Compound **17** showed ion at *m*/*z* 797.4688 with a molecular formula C_42_H_69_O_14_. It was identified as a Bacopaside N, as the MS^2^ spectra showed losses of two sugar units from the jujubogenin aglycon [35].

### 3.2. ST-65 Viability

To study the biological effect of ST-65, we preliminary evaluated the extract’s viability effect on SH-SY5Y neuroblastoma cells. Figure 2 clearly shows that the ST-65 extract did not significantly affect the viability of cells after 24 h of treatment, even at the highest concentrations.

### 3.3. NMR Metabolomic

The cytoplasmatic metabolites, produced by cells (endometabolome) and the extracytoplasmatic metabolites (esometabolome) were investigated using ^1^H-NMR spectroscopy. We performed an untargeted metabolomic analysis on SH-SY-5Y cell cultures treated and untreated with the Bacopa and Ashwagandha formulations called ST-65. Figure 3A,B shows the representative 1D ^1^H NOESY NMR spectrum of cellular endo and esometabolome, respectively [36]. The spectra were assigned using Chenomx NMR-Suite v8.0 [37], highlighting 41 metabolites in the endometabolome and 36 in the esometabolome. NMRProcFlow was used to quantify the spectra, taking into account the internal standard trimethylsilyl propanoic acid (TSP), the number of H for each peak, and the metabolites’ molecular weight [38]. After quantification, metabolites matrices were analysed using MetaboAnalyst 6.0 by univariate and multivariate supervised approaches.

The dataset was first normalised by Log and Parate scaling to perform Partial Least Square determination analysis (PLS-DA) [39]. The score plots in Figure 4A,B showed a different metabolomic profile between the endo metabolome and the esometabolome of treated and untreated cells. The score plot was described in the Cartesian space by the main components: PC1: 68.6% and PC2:17.3% for endometabolome. On the other hand, the esometabolome score plot was described as PC1: 65.2% and PC2:30.1%. The clusters’ separation was confirmed by cross-validation. (Appendix A) [40]. The metabolites responsible for clusters’ separation were identified according to the Variable Importance Projections (VIP) score analysis [41]. Accordingly, the metabolites with a VIP score > 1.00 are considered good classifiers between the clusters. (Figure 4C,D). VIP analysis showed a dysregulation of several energetic metabolites: in particular, cells treated with ST-65 produced lower concentrations of Formate and 2-oxobutyrate related to cholesterol pathways and Acetate, and Carnitine. The analysis identified the upregulation of several metabolites related to mitochondria, such as ADP, AMP, and Coenzyme A. (Figure 4B) ST-65 modulated aminoacidic metabolism; in particular, we detected a downregulated production of cysteine and aspartate. Conversely, the endometabolome aminoacidic concentration of proline, tyrosine, leucine and glutamate was higher in the cells treated with ST-65 and lower in the untreated cells. Moreover, VIP score analysis detected higher concentrations of tryptophan, tyrosine and valine in the esometabolome of treated cells and lower of arginine and carnitine (Figure 4D). The volcano plot confirmed the results of the VIP score analysis. Additionally, it showed a higher concentration of phenylalanine and a lower concentration of serine and glucose in the endomebolome of SH-SY-5Y treated cells (Figure 4E). Enrichment pathways analysis was performed to identify the dysregulated pathways according to the metabolites ranked by VIP score analysis.

The paths with hits > 1 and *p* value < 0.05 were considered discriminatory [33] (Figure 5; Appendix A). Enrichment revealed that ST-65 impacts several energetic pathways, such as Pyruvate metabolism and the Citric acid Cycle. Furthermore, mitochondrial biochemical pathways related to fatty acids, such as beta-oxidation of short-chain fatty acids in mitochondria, beta-oxidation of long-chain fatty acids, and ketosis body metabolism, are also impacted. Moreover, ST-65 also modulates pathways related to steroid hormone biosynthesis, derived from cholesterol and regulated by gonadotropins, and neurotransmitter amino acids, such as glycine and serine metabolism.

## 4. Discussion

The combination of ashwagandha (*Withania somnifera*) and bacopa (*Bacopa monnieri*) has garnered attention in clinical settings due to their individual and potentially synergistic therapeutic effects. Both herbs have a long history of use in traditional medicine, and emerging clinical evidence supports their benefits for stress management, cognitive enhancement, and overall health and their combination may offer a synergistic effect due to their complementary pharmacological actions [7,8,9,10]. While ashwagandha is known for its adaptogenic properties, bacopa is recognized for cognitive enhancement. The optimization of UHPLC conditions achieved maximal chromatographic resolution and MS signal was crucial for accurately identify the bioactive compounds present in the in the ashwagandha-bacopa formulation. The chemical profile of ST-65 formulation is related to the presence of the extract of *Withania somnifera* and *Bacopa monnieri.* Withanolides are marker compounds of the Solanaceae, especially in the *Withania* genus [16,25]. The dammarane-type triterpenoid saponin known as bacoside, are characterized phytochemicals in *Bacopa Monnieri* [25].

Withanolides are known for their adaptogenic properties, and their presence may explain some of the formulation’s effects on stress and cognitive function, this compounds may help alleviate these conditions mainly by modulating the hypothalamic-pituitary-adrenal and sympathetic-adrenal medullary axes, as well as through the GABAergic and serotonergic pathways [10]. The detection of a bacopa saponin (peak 17) is another critical finding. *Bacopa monnieri* is traditionally used for enhancing cognitive function, and its saponins are believed to be responsible for its neuroprotective effects, the neuroprotective activity of bacosides is linked to their role in regulating mRNA translation and the surface expression of neuroreceptors, including AMPAR, NMDAR, and GABAR, in different regions of the brain [24,27]. The metabolomic approach performed on cell cultures allowed us to identify a different metabolomic profile of ST-65 treated cells compared with untreated cells, showing a clear clustering of the two groups following a multivariate PLS-DA approach (Figure 4A). SH-SY5Y cells were treated with ST-65 to evaluate its action, mainly on neuronal energy. The metabolomics analysis validated this hypothesis and revealed further modulated mechanisms. SH-SY5Y cells treated with ST-65 showed modulation of aminoacidic pathways involved in neurotransmission. In detail, VIP analysis highlighted an increase in the concentrations of glutamate and a decrease in the intracellular concentrations of aspartate. At the same time, a lower tryptophan use was identified, accompanied by an increased production and expulsion of tyrosine (Figure 4C,D). These results suggest that ST-65 could influence the metabolism of amino acids participating in neurotransmission and neuroplasticity processes [42]. Glutamate is also essential as it participates in cellular energy and is the most abundant free amino acid in the brain [43]. The enrichment analysis identified among the most modulated pathways in cells treated with ST-65 pyruvate metabolism and citric acid cycle (Figure 5). The increase in glutamate induced by treatment with ST-65 participates in the upregulation of the citric acid cycle. It also leads to a greater use of ATP, as demonstrated by the increase in its AMP and ADP products. The variations of ADP suggest that ST-65 could modulate neuronal energy positively. It is known that ADP stimulates respiration through oxidation processes, which in turn provide energy [44]. The components of ST-65, especially Ashwagandha, are known to improve the production of sex hormones [45]. Our findings support this effect, demonstrating a clear enhancement in hormone production. The treatment with STRESSLESS (ST-65) impacted the biosynthesis of steroids and sex hormones, as demonstrated by the enrichment analysis (Figure 5), which identified among the most dysregulated pathways: steroid biosynthesis and androgen and estrogen metabolism. Estrogens and androgens exhibit antioxidative properties, mitigating oxidative stress within the brain—a critical factor implicated in neurodegeneration. These hormones inhibit neuronal apoptosis by modulating key signaling pathways, including MAPK and PI3K/Akt. Additionally, they promote synaptic plasticity and neurogenesis, thereby enhancing cognitive processes such as learning and memory. Furthermore, sex hormones play a role in immunomodulation by regulating cytokine release [15]. It is known that the synthesis of sex hormones is closely related to cholesterol pathways [46]. Our study has shown a decrease in 2-oxobutyrate and intracellular formate (Figure 4C) in cells treated with ST-65. Formate is produced by the synthesis of cholesterol and by the degradation of folate deficiency [47]. A reduction proves an improvement of these biochemical pathways, indicating that ST-65 could also have a lipid-lowering action. Pathway analysis supports this hypothesis by identifying biochemical pathways related to the oxidation of short- and long-chain fatty acids among the dysregulated pathways. (Figure 5) As is known, the processes of beta-oxidation of fatty acids occur at the mitochondrial level. Our study identified a dysregulation of the biochemical pathways of mitochondrial oxidation in ST-65 -treated cells and a consequent increase of Coenzyme A concentration (Figure 4C), a metabolite essential for cellular bioenergetics, especially for processes related to lipid metabolism [48], the oxidation of fatty acids is crucial in the myelinating cells of the CNS, specifically oligodendrocytes, where it contributes to lipid turnover and supplies essential components for the synthesis of new myelin lipids [49,50]. These findings contribute to the understanding of protective mechanisms for neurodegenerative diseases (NDDs) by highlighting the role of Coenzyme A in supporting cellular energy production and stability under stress [51]. CoA plays a pivotal role in neuroprotection by enabling ATP synthesis essential for neuronal function, mitigating oxidative stress through ROS management, enhancing mitochondrial dynamics (e.g., fission, fusion, and biogenesis), and facilitating the clearance of damaged mitochondria to maintain cellular health [52]. Additionally, CoA influences gene expression via acetylation, regulates neuroinflammation, and supports neurosteroid synthesis, all of which are critical for neuronal survival [53]. This aligns with existing theories on the importance of mitochondrial function in neuroprotection, particularly in diseases such as Alzheimer’s and Huntington’s, and reinforces prior experimental evidence on bioenergetic pathways in cellular resilience. For instance, Sang et al. examined six postmortem brain regions affected by Alzheimer’s disease and identified significant perturbations in mitochondrial TCA cycle enzymes, such as pyruvate dehydrogenase complex, isocitrate dehydrogenase, 2-oxoglutarate dehydrogenase complex, and succinyl-CoA synthetase, all of which are directly or indirectly regulated by CoA [54]. By connecting these findings to established research, our work underscores the critical role of CoA in neurodegenerative disease pathology and highlights its potential as a target for therapeutic intervention.

Metabolomics analysis also identified an increase in cell uptake and metabolism of carnitine by ST-65-treated cells (Figure 4C,D). Carnitine is a central metabolite in the peroxidation process of fatty acids and in the biochemical processes of free radical scavengers [55]. Carnitine, widely known for its significant role in transporting LCFAs across the inner mitochondrial membrane, is essential for proper neurological function [56]. Although fat is not the primary fuel for the brain, supplementation with acylcarnitine and carnitine has shown positive effects in treating various neurological disorders [57]. Recent research suggests that acylcarnitines play multifaceted roles in neuroprotection. Brain acylcarnitines are involved in lipid synthesis, modulation and stabilization of membrane composition, regulation of genes and proteins, enhancement of mitochondrial function, increased antioxidant activity, and improvement of cholinergic neurotransmission [58]. Given these roles, carnitine and acylcarnitines are increasingly recognized for their potential to mitigate the effects of neurological disorders, such as Alzheimer’s disease, Parkinson’s disease, depression, and peripheral neuropathy. In conclusion, the metabolomics analysis showed that ST-65 affects energy metabolism, particularly having a hypolipidemizing and hypoglycemic action. It also assists the synthesis of sex hormones and improves neuronal transmission and mitochondrial function. Further investigations are warranted to elucidate the specific pathways involved and to evaluate its clinical applicability.

## 5. Conclusions

This study highlights the potential therapeutic benefits of the combination of ashwa-gandha (*Withania somnifera*) 600 mg and bacopa (*Bacopa monnieri*) 50 mg (STRESSLESS or ST-65 formulation) through a multifaceted metabolomic approach. The results provide compelling evidence for the synergistic effects of these two herbs, particularly in stress management, cognitive enhancement, and neuroprotection. Key bioactive com-pounds, including withanolides from ashwagandha and saponins from bacopa, were identified as potentially responsible for modulating stress-related pathways and enhancing cognitive function. The metabolomics analysis revealed that the ST-65 significantly enhances mitochondrial bioenergetics by increasing glutamate levels and modulating the citric acid cycle, as well as by upregulating Coenzyme A (CoA) levels. Mitochondrial dysfunction is a hallmark of NDDs such as Alzheimer’s and Huntington’s diseases, and the observed improvements in mitochondrial dynamics and energy production suggest potential protective effects against these conditions. The increased uptake of carnitine and modulation of fatty acid oxidation pathways indicate that ST-65 could mitigate oxidative stress, a critical driver of neurodegeneration. The ability of ST-65 to promote lipid metabolism is particularly relevant for myelination processes and the maintenance of oligodendrocyte function in diseases like multiple sclerosis. Moreover the observed upregulation of steroid biosynthesis pathways, including androgen and estrogen metabolism, highlights ST-65’s role in modulating neurosteroids. These hormones not only mitigate oxidative stress but also promote neurogenesis, synaptic plasticity, and immunomodulation, critical for managing NDD progression. Related to neuroprotection mechanisms, the metabolomic analysis revealed that ST-65 modulates amino acid pathways, including increases in glutamate and reductions in aspartate and tryptophan use. Glutamate’s role in cellular energy and neurotransmis-sion, combined with its regulated activity in the TCA cycle, supports neuronal resilience and plasticity, both crucial for neuroprotection. Enrichment analysis showed the involvement of pathways regulating oxidative stress and apoptosis, such as MAPK and PI3K/Akt. These findings align with the known protective effects of adaptogens and neurosteroids in reducing neuroinflammation and neuronal death. Our results suggest potential applications of STRESSLESS in managing stress, early-stage cognitive decline, mild cognitive impairment, and neurodegenerative conditions. The adaptogenic properties of ashwagandha and cognitive benefits of bacopa provide a strong basis for further exploration in human clinical trials.

## Figures and Tables

**Figure 1 nutrients-16-04096-f001:**
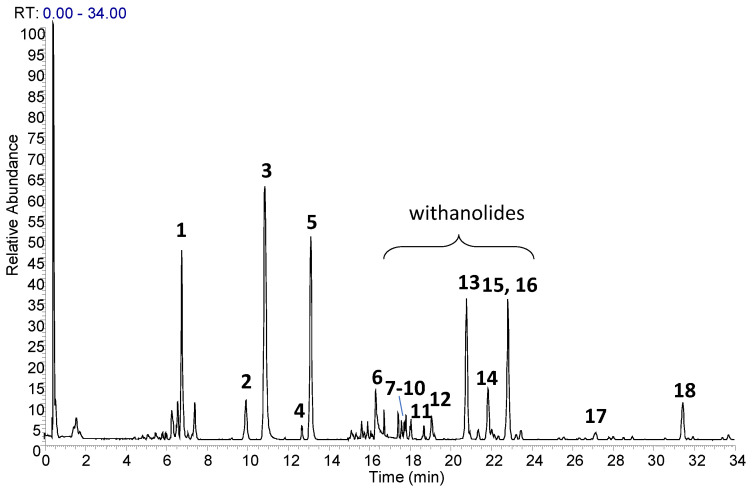
UHPLC-HRMS profile of ST-65 Formulation.

**Figure 2 nutrients-16-04096-f002:**
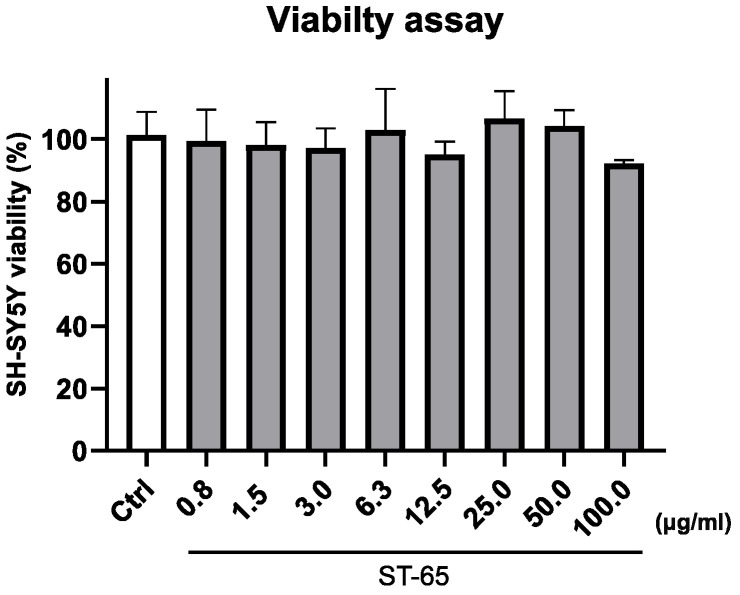
Histogram graph showing the percentage of cell viability 24 h after the exposition of different concentrations of ST-65 extract (0.80–100 µg/mL). The viability variations were calculated as the percentage of viable cells in treated cultures compared to untreated ones. Results are shown as mean ± standard deviation (SD) from three independent experiments. The statistical significancy was evaluated using the one-way analysis of variance (ANOVA), followed by Bonferroni’s test, with GraphPad Prism 8.0 software (San Diego, CA, USA) assuming significance at *p* < 0.05. The statistical test showed the viability of cell exposed to each concentration of ST-65 is not significantly different from the control.

**Figure 3 nutrients-16-04096-f003:**
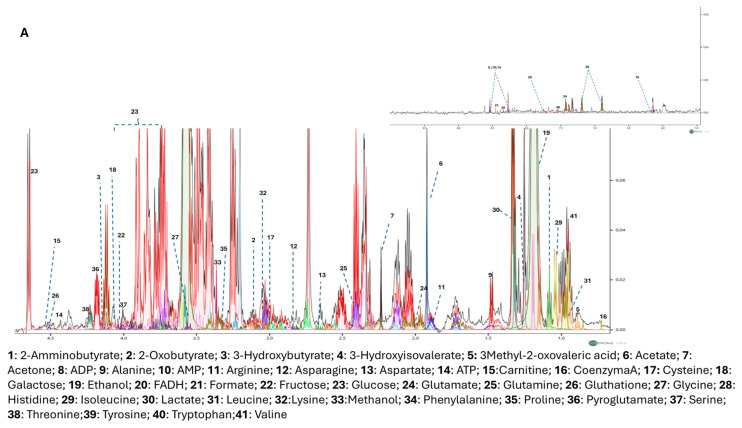
1D shows the NOESY spectrum of the endometabolome (**A**) and esometabolome (**B**) of SH-SY-5Y polar extracts. The spectrum was acquired, as mentioned before. Forty-one and Thirty-six metabolites were detected in the endometabolome and esometabolome, respectively.

**Figure 4 nutrients-16-04096-f004:**
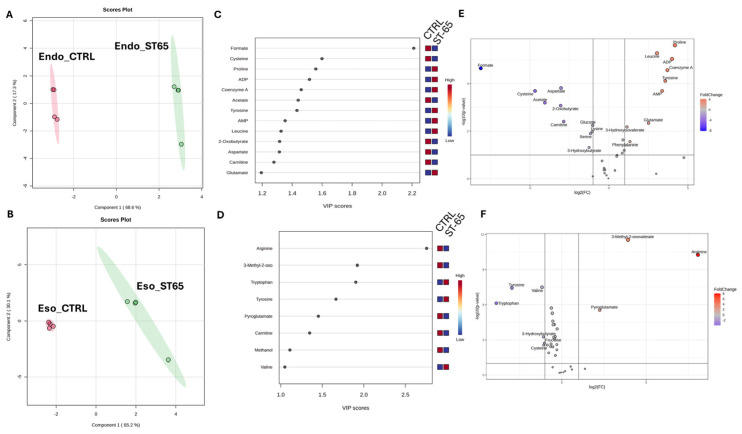
PLS-DA score plots related to the endometabolome (**A**) and the esometabolome (**B**) of treated (green) and untreated (red) cells with ST-65 formulation. VIP score graph reported the metabolites responsible for clusters endometabolome (**C**) and esometabolome cluster separation (**D**). Volcano plot analysis of metabolic changes in the cellular polar extract of endo (**E**) and esometabolome (**F**) of treated and untreated cells. Each point on the volcano plot was based on *p*-value and fold-change values, set at 0.05 and 2.0, respectively. Red points identify upregulated metabolites, whereas blue points identify down-regulated metabolites.

**Figure 5 nutrients-16-04096-f005:**
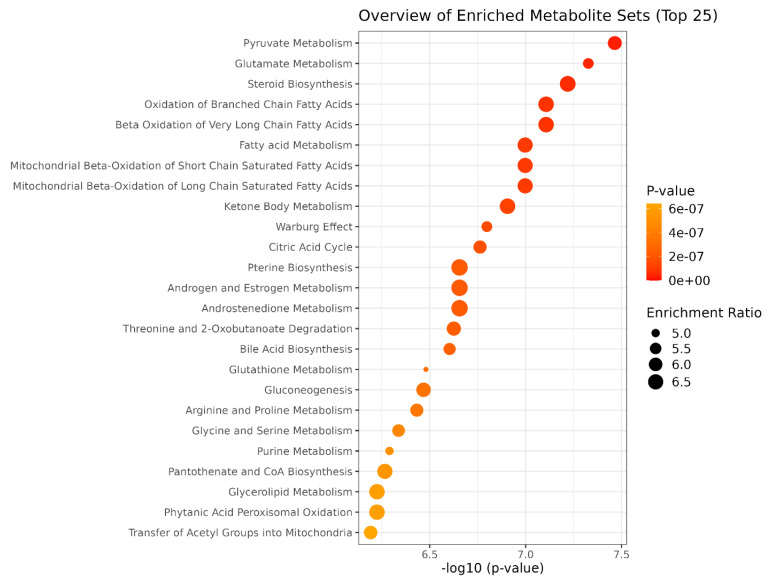
Enrichment pathways analysis. The y-axis shows the discriminating metabolic pathways in order of *p*-value (expressed on the x-axis as negative of logarithm) increasing. The size of the circles represents the number of hits given as an enrichment ratio.

**Table 1 nutrients-16-04096-t001:** UHPLC-HRMS data of compounds detected in ST-65 formulation.

N. ^a^	RT [min]	*m*/*z*	Formula	ppm	MS/MS	Name
1	6.8	224.1644	C_13_H_22_O_2_N	−0.47	124, 93	unknown
2	9.9	226.1801	C_13_H_24_O_2_N	−0.334	144, 126, 84	unknown
3	10.9	611.1597	C_27_H_31_O_16_	−1.524	465, 303	Rutin ^b^
4	12.71	595.1658	C_27_H_31_O_15_	0.241	449, 287	Kaempferol hexose-deoxyhexose
5	13.1	611.1959	C_28_H_35_O_15_	−1.745	447, 303	Hesperidin ^b^
6	16.3	303.0497	C_15_H_11_O_7_	−0.723	153	Quercetin ^b^
7	17.4	783.4157	C_40_H_63_O_15_	−0.482	459, 441, 423	Withanoside IV/Withanoside X
8	17.6	489.2847	C_28_H_41_O_7_	0.225	317, 299, 281	Vicosalactone B isomer
9	17.8	489.2847	C_28_H_41_O_7_	0.163	317, 299, 281	Vicosalactone B isomer
10	17.8	783.4156	C_40_H_63_O_15_	−0.635	459, 441, 423	Withanoside IV/Withanoside X
11	18.1	621.363	C_34_H_53_O_10_	−0.457	459, 441, 423	Coagulin Q
12	18.7	621.3632	C_34_H_53_O_10_	−0.264	459, 441, 423	Coagulin Q isomer
13	20.8	471.2737	C_28_H_39_O_6_	−0.733	435, 341, 299, 281	Whitanolide A isomer
14	21.8	471.2739	C_28_H_39_O_6_	−0.457	435, 341, 299, 281	Whitanolide A isomer
15	22.8	471.2737	C_28_H_39_O_6_	−0.797	435, 341, 299, 281	Whitanolide A isomer
16	22.8	767.4209	C_40_H_63_O_14_	−0.434	443, 425, 407, 389	Withanoside V
17	28.0	797.4688	C_42_H_69_O_14_	0.861	599, 441, 423	Bacopaside N
18	31.4	931.5315	C_40_H_83_O_23_	−0.51	477	unknown

^a^ Compounds are numbered according to their elution order. ^b^ Compared with reference standards.

## Data Availability

Data are contained within the article and Appendix A.

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
