# Peer review of "Impact of a Withania somnifera and Bacopa monnieri Formulation on SH-SY5Y Human Neuroblastoma Cells Metabolism Through NMR Metabolomic"

_nutrients, 2024, doi:10.3390/nu16234096_

Round 1

Reviewer 1 Report

Comments and Suggestions for Authors

In the present research work authors have shown the potential impact of a Withania somnifera and Bacopa monnieri Formulation on SH-SY5Y Human Neuroblastoma Cells Metabolism through NMR Metabolomic, a possible therapeutic potential for treatment of NDDS and in neuroprotection.

The article is planned well however it lacks major therapeutic impact of Withania somnifera and Bacopa monnieri Formulation on neuroprotection and NDDs treatment and management.

Article lacks the impact of their work by connecting their findings more explicitly to neurodegeneration and neuroprotective mechanisms, which are crucial for contextualizing the study within the field.

Minor suggestions

1.      Improve the presentation of Figure 2.

2.      Increase the font size of X and Y- axis of Figure 3.

3.      Enhance the resolution of Figure 4.

4.      line 319, give space between amino acid.

5.      Line 327, delete energy (Maldonado & Lemas ters, 2014).

6.      Line 329, Our results confirmed this action, is not appropriator. Rewrite the sentence.

Major comments

1.      Lines 329-332, discussed about sex hormone metabolism however in introduction and results authors given any major impact of their study on sex hormone metabolism and hows this corroborated with neuroprotection.

2.      The discussion on lines 340-344 should focus on the findings' contribution to understanding protective mechanisms for neurodegenerative diseases (NDDs) and their relevance to neuroprotection. It should also connect these findings to existing theories and experimental evidence, demonstrating their relevance and alignment with established research.

3.      lines 345-347, discussed this citation work with present study results.

4.      While the authors mention in the abstract that their findings relate to "stress-related neurodegenerative conditions" and neuroprotection, they do not provide detailed analysis or discussion of these aspects in the results or discussion sections.

5.      The experimental work is well-executed; however, the authors have not effectively demonstrated the potential applications of their findings for neurodegenerative diseases and neuroprotection. Expanding on the relevance and translational impact of their results in these areas would strengthen the study's significance.

6.      The conclusion lacks a clear summary of the study's key findings, advantages, and limitations, requiring a stronger discussion to provide a comprehensive understanding of its impact.

Author Response

In the present research work authors have shown the potential impact of a Withania somnifera and Bacopa monnieri Formulation on SH-SY5Y Human Neuroblastoma Cells Metabolism through NMR Metabolomic, a possible therapeutic potential for treatment of NDDS and in neuroprotection.

The article is planned well however it lacks major therapeutic impact of Withania somnifera and Bacopa monnieri Formulation on neuroprotection and NDDs treatment and management.

Comment 1: Article lacks the impact of their work by connecting their findings more explicitly to neurodegeneration and neuroprotective mechanisms, which are crucial for contextualizing the study within the field.

Response 1: We believe the revisions address the reviewer’s concerns and strengthen the translational impact of our study. We thank the reviewer for their valuable input, which has enabled us to improve the clarity and significance of our manuscript.

Minor suggestions

Comment 2: Improve the presentation of Figure 2.

Response 2: We improved the aspect of the graph and the axis labels. Moreover, we included in the histogram the viability of the control, treated only with vehicle.

Comment 3: Increase the font size of X and Y- axis of Figure 3.

Response 3: We improved size of X and Y- axis of Figure

Comment 4: Enhance the resolution of Figure 4.

Response 4: We enhanced the resolution of Figure 4.

Comment 5: line 319, give space between amino acid.

Response 5: Done

Comment 6: Line 327, delete energy (Maldonado & Lemas ters, 2014).

Response 6: Done

Comment 7: Line 329, Our results confirmed this action, is not appropriator. Rewrite the sentence.

Response 7: Done, we have rewritten the sentence

Major comments

Comment 8: Lines 329-332, discussed about sex hormone metabolism however in introduction and results authors given any major impact of their study on sex hormone metabolism and how this corroborated with neuroprotection.

Response 8: We discussed the impact of sex hormone metabolism on neuroprotection in introduction and results. On lines 58-68 of introduction we give a scientific overview of how sex hormones, particularly estrogen, progesterone, and testosterone, contribute to neuroprotection. Also in the discussion session we included at 357-362 the protective role against neurodegeneration of estrogens and androgens

Comment 9: The discussion on lines 340-344 should focus on the findings' contribution to understanding protective mechanisms for neurodegenerative diseases (NDDs) and their relevance to neuroprotection. It should also connect these findings to existing theories and experimental evidence, demonstrating their relevance and alignment with established research.

Response 9: On lines 375-392 we addressed the request by emphasizing how our findings contribute to the understanding of neurodegenerative diseases and their protective mechanisms. we explained the role of Coenzyme A (CoA) in neuroprotection, including its impact on ATP production, ROS management, mitochondrial dynamics, and cellular health. We articulated the alignment with established theories and the context of specific neurodegenerative diseases like Alzheimer's and Huntington's. By discussing perturbations in enzymes regulated by CoA, we demonstrated a connection to broader bioenergetic pathways in neurodegenerative diseases. We compared our findings with relevant literature or previous studies, We cited the study of Sang et al. which provides experimental evidence supporting our findings. This expands the understanding of CoA's multifaceted role in neuroprotection and suggests its potential as a broader therapeutic target across multiple neurodegenerative diseases.

Comment 10: lines 345-347, discussed this citation work with present study results.

Response 10: On lines 395-404 we discussed the role of carnitine in energy metabolism, neuroprotection, and modulation of neurochemical pathways. We cited 3 new articles (57-59) reporting its  therapeutic potential in managing and alleviating neurological disorders. Given these roles, carnitine and acylcarnitines are increasingly recognized for their potential to mitigate the effects of neurological disorders, such as Alzheimer's disease, Parkinson's disease, depression, and peripheral neuropathy. The described functions align with our findings from metabolomics analyses that suggest increased cellular uptake and metabolism of carnitine in treated cells

Comment 11: While the authors mention in the abstract that their findings relate to "stress-related neurodegenerative conditions" and neuroprotection, they do not provide detailed analysis or discussion of these aspects in the results or discussion sections.

Response 11: We replied discussing the  the impact on sex hormone and the role of CoA as reported before

Comment 12: The experimental work is well-executed; however, the authors have not effectively demonstrated the potential applications of their findings for neurodegenerative diseases and neuroprotection. Expanding on the relevance and translational impact of their results in these areas would strengthen the study's significance.

Response 12: We thank the reviewer for their thoughtful comments, which have prompted us to expand on the relevance and translational impact of our findings for neurodegenerative diseases (NDDs) and neuroprotection. We have expanded the Discussion section to detail these connections, linking our findings to specific neurodegenerative conditions and emphasizing their therapeutic relevance. We have included additional citations to align our findings with established research in the field of neuroprotection and NDDs. Our study demonstrated that ST-65 enhances mitochondrial bioenergetics by increasing glutamate levels and modulating the citric acid cycle, as well as by upregulating Coenzyme A (CoA) levels. Mitochondrial dysfunction is a hallmark of NDDs such as Alzheimer's and Huntington's diseases, and the observed improvements in mitochondrial dynamics and energy production suggest potential protective effects against these conditions.

Comment 13: The conclusion lacks a clear summary of the study's key findings, advantages, and limitations, requiring a stronger discussion to provide a comprehensive understanding of its impact.

Response 13: In this revised version we  offered in the conclusion a concise and structured summary that clearly communicates the study’s significance, the advantages of our formulation, and its limitations.

Reviewer 2 Report

Comments and Suggestions for Authors

The study aims to investigate the neuroprotective effects of a combination of Withania somnifera and Bacopa monnieri using human neuroblastoma cells, based on the assessment of changes in the metabolic profile after treatment with the formulation. The manuscript is interesting and provides new findings that could justify the application of both herbs in traditional medicine for stress alleviation. Overall, this is a well-conducted study, and I have only a few suggestions for the authors.

Introduction: Please add a few  information on the biological activity of withanolides. The section about Bacopa monnieri should start as a new paragraph.

Line 73-78: „In our nutrition research laboratory (Nutriketo Lab) we have been involved in conducting dietary clinical trials to explores the ketogenic diet's applications in various medical contexts…….” – In my opinion, this section is unnecessary unless the cited studies specifically link the ketogenic diet with Withania somnifera or Bacopa monnieri preparations. If so, this connection should be clearly stated. Otherwise, this section appears to serve only as a way for the authors to cite their own papers.

Line 104: „1200 mg of Ashwagandha E.S. and 100 mg of Bacopa” – Why did the authors use this particular proportion? Please add an explanation in the text.

Line 111: „A Kinetex C18 111 column (…..) was used a flow rate of 500 μL min-1 at 30 °C.” – correct the sentence

Line 113: „Volume of the injection was 5 μL.” – unnecessary. This information is in line 117

Line 120: „Full MS scan resolution (scan range m/z 120 100-1000), 60000 FWHM; dd-MS2 scan set, 30000 FWHM.” – correct the sentence (lack of verb)

Line 137: „(8 × 103 cells/well)” - correct

Line 138: „ and then ST -65 in the range of 100-0.80 μg/mL was  added for an additional 24 h.” – The range should be presented from the lowest value to the highest.

„2.6. Exposure of SH-SY5Y to ST-65”  - Add “for metabolite investigation” (or something similar) for better clarity.

Line 146: „(25 'g/ml)”  - correct unit

2.9. Statistical analysis – The information in this section should be moved to the section on NMR analysis. Typically, the statistical analysis section includes the number of replications and the tests used to assess the significance of differences.

Figure 1: The chromatogram lacks a baseline. Please provide another version. The figure legend should  be more detailed.

Figure 2: Data should be presented from the lowest to the highest concentration. Are the data statistically significantly different from the control?

Figures 3 and 4: The quality of these figures is very poor, and the descriptions below the figures are unreadable.

In the Discussion section, the authors should compare their findings on the phytochemical profile with existing data on the chemical composition of the investigated plants.

Line 327: remove: „(Maldonado & Lemasters, 2014)”

Author Response

Comment 1: The study aims to investigate the neuroprotective effects of a combination of Withania somnifera and Bacopa monnieri using human neuroblastoma cells, based on the assessment of changes in the metabolic profile after treatment with the formulation. The manuscript is interesting and provides new findings that could justify the application of both herbs in traditional medicine for stress alleviation. Overall, this is a well-conducted study, and I have only a few suggestions for the authors.

Response 1: We thank the reviewer for their thoughtful comments, we added few  information on the biological activity of withanolides.

Comment 2: Line 73-78: „In our nutrition research laboratory (Nutriketo Lab) we have been involved in conducting dietary clinical trials to explores the ketogenic diet's applications in various medical contexts…….” – In my opinion, this section is unnecessary unless the cited studies specifically link the ketogenic diet with Withania somnifera or Bacopa monnieri preparations. If so, this connection should be clearly stated. Otherwise, this section appears to serve only as a way for the authors to cite their own papers.

Response 2: we followed the referee's suggestions and removed this part and the linked bibliography

Comment 3: Line 104: „1200 mg of Ashwagandha E.S. and 100 mg of Bacopa” – Why did the authors use this particular proportion? Please add an explanation in the text.

Response 3: We clarified this point in the introduction lines 91-95

Comment 4: Line 111: „A Kinetex C18 111 column (…..) was used a flow rate of 500 μL min-1 at 30 °C.” – correct the sentence

Response 4: We corrected

Comment 5: Line 113: „Volume of the injection was 5 μL.” – unnecessary. This information is in line 117

Response 5: We deleted this information

Comment 6: Line 120: „Full MS scan resolution (scan range m/z 120 100-1000), 60000 FWHM; dd-MS2 scan set, 30000 FWHM.” – correct the sentence (lack of verb)

Response 6: We corrected

Comment 7: Line 138: „ and then ST -65 in the range of 100-0.80 μg/mL was  added for an additional 24 h.” – The range should be presented from the lowest value to the highest.

Response 7: done

Comment 8: „2.6. Exposure of SH-SY5Y to ST-65”  - Add “for metabolite investigation” (or something similar) for better clarity.

Response  8: Done we changed in “Exposure of SH-SY5Y to Stressless (ST-65) formulation”

Comment 9: Line 146: „(25 'g/ml)”  - correct unit

Response  9: We corrected

Comment 10: 2.9. Statistical analysis – The information in this section should be moved to the section on NMR analysis. Typically, the statistical analysis section includes the number of replications and the tests used to assess the significance of differences.

Response 10: Done. We moved the section

Comment 11: Figure 1: The chromatogram lacks a baseline. Please provide another version. The figure legend should  be more detailed.

Response 11:We provided

Comment 12: Figure 2: Data should be presented from the lowest to the highest concentration. Are the data statistically significantly different from the control?

Response 12: Thank you for the comment. The statistical significancy was evaluated using the one-way analysis of variance (ANOVA), followed by Bonferroni’s test, with GraphPad Prism 8.0 software (San Diego, CA, USA) assuming significance at p<0.05. The statistical test showed the viability of cell exposed to each concentration of ST-65 is not significantly different from the control. Moreover, we improved the graph including the control (treated only with vehicle) viability.

Comment 13: Figures 3 and 4: The quality of these figures is very poor, and the descriptions below the figures are unreadable.

Response 13: Done we improved figures 3 and 4

Comment 14: In the Discussion section, the authors should compare their findings on the phytochemical profile with existing data on the chemical composition of the investigated plants.

Response 14: Thank you for the comment , we reported it on lines 318-320

Comment 15: Line 327: remove: „(Maldonado & Lemasters, 2014)”

Response 15: removed

Round 2

Reviewer 1 Report

Comments and Suggestions for Authors

The authors have successfully addressed all the issues raised in the previous review. Their responses are satisfactory, and the relevant changes have significantly enhanced the overall quality of the manuscript. The manuscript now meets the required standards and can be accepted in its current form.